# Nanomedicine for Treating Muscle Dystrophies: Opportunities, Challenges, and Future Perspectives

**DOI:** 10.3390/ijms231912039

**Published:** 2022-10-10

**Authors:** Zaheer Ahmed, Rizwan Qaisar

**Affiliations:** 1Institute for Experimental Molecular Imaging, RWTH Aachen University Hospital, 52074 Aachen, Germany; 2Department of Basic Medical Sciences, College of Medicine, University of Sharjah, Sharjah 27272, United Arab Emirates; 3Cardiovascular Research Group, Sharjah Institute for Medical Research, University of Sharjah, Sharjah 27272, United Arab Emirates

**Keywords:** nanoparticles, skeletal muscle, dystrophies, small molecules

## Abstract

Muscular dystrophies are a group of genetic muscular diseases characterized by impaired muscle regeneration, which leads to pathological inflammation that drives muscle wasting and eventually results in weakness, functional dependency, and premature death. The most known causes of death include respiratory muscle failure due to diaphragm muscle decay. There is no definitive treatment for muscular dystrophies, and conventional therapies aim to ameliorate muscle wasting by promoting physiological muscle regeneration and growth. However, their effects on muscle function remain limited, illustrating the requirement for major advancements in novel approaches to treatments, such as nanomedicine. Nanomedicine is a rapidly evolving field that seeks to optimize drug delivery to target tissues by merging pharmaceutical and biomedical sciences. However, the therapeutic potential of nanomedicine in muscular dystrophies is poorly understood. This review highlights recent work in the application of nanomedicine in treating muscular dystrophies. First, we discuss the history and applications of nanomedicine from a broader perspective. Second, we address the use of nanoparticles for drug delivery, gene regulation, and editing to target Duchenne muscular dystrophy and myotonic dystrophy. Next, we highlight the potential hindrances and limitations of using nanomedicine in the context of cell culture and animal models. Finally, the future perspectives for using nanomedicine in clinics are summarized with relevance to muscular dystrophies.

## 1. Introduction

Muscular dystrophies are a group of genetic muscular diseases characterized by progressive muscle wasting and weakness that lead to functional dependency, disability, and premature death [1]. Altogether, muscular dystrophies involve mutations in over 40 genes with distinct pathogenic mechanisms and clinical presentations [2]. The global estimated prevalence of muscular dystrophies is 19.8—25.1 per 100,000 births [2]. The two most common forms of muscular dystrophy are Duchenne muscular dystrophy (DMD) and myotonic dystrophy (DM) [1]. DMD is caused by a mutation in the dystrophin gene, which results in sarcolemma damage during contractile activity [3]. DM involves mutations in drug metabolism and pharmacokinetics (DMPK) or cellular nucleic acid-binding protein (CNBP) genes, which result in dysregulated RNA splicing and contractile dysfunction [4]. These conditions lead to repeated myofibers injuries during routine contractile activities [1]. After exhausting its regenerative capacity, the skeletal muscle develops myofiber atrophy, progressive fibrosis, infiltration of inflammatory cells, and neuromuscular junction degeneration [5]. The clinical presentation involves progressive muscle atrophy and weakness, which leads to premature death due to dysfunction of respiratory muscles [1].

Currently, therapeutic interventions do not have the efficacy required to inhibit or reverse the pathological endpoints of muscular dystrophies. The management is primarily focused on reducing the complications and symptoms of the disease to improve the quality of life [1]. Corticosteroids are a standard treatment option but demonstrate limited efficacy and present long-term adverse effects [6]. Exercise may be a therapeutic option in muscular dystrophy but show poor compliance and may not be possible in patients with functional disabilities [7]. Recently, small molecules and novel gene therapies have been proposed as experimental interventions, but their adverse effects and rapid clearance from the body prevent promising clinical trials [8].

Over the last several years, nanomedicine has emerged as an attractive therapeutic option for many human diseases [9]. Nanoparticles have been successfully used to deliver oligonucleotides [10], cytokines [11], and other bioactive molecules, including growth factors [2], to promote the regeneration of target organs. Currently, several nanoparticles are being optimized to boost skeletal muscle mass and/or strength in different muscle-wasting conditions, including muscular dystrophies [2,8]. However, several barriers prevent the effective delivery of nanoparticles to dystrophic skeletal muscles. These include an abundance of fibrous tissues, inadequate optimization of delivery systems, and loss of bioactivity of the therapeutic molecules [1,4,5].

The present review summarizes the major advances in the therapeutic applications of nanomedicine in muscular dystrophies. We have discussed the advances and limitations of various treatment strategies based on nanomedicine. Finally, we summarize the future perspective of nanomedicine in treating muscular dystrophies. 

## 2. Nanomedicine, Its History, and Applications

Nanomedicine is a rapidly evolving interdisciplinary field that stems from the convergence of nanotechnology and medicine. It seeks to manipulate and manufacture new functional materials and devices in size range of 1–100 nm for the diagnosis, monitoring, control, prevention, and treatment of various diseases [12]. 

Recent advancement in nanotechnology has led to the development of various nanoparticle formulations for therapeutic and diagnostic applications [13]. In a diagnostic setting, nanoparticles assist in understanding the pathophysiology of various diseases. In contrast, therapeutically, nanoparticles deliver and accumulate the payload at the pathological sites, which increases therapeutic efficacy and reduces off-target effects [14].

The advanced drug delivery carriers can be organic or inorganic based on their chemical compositions. Organic nanoparticles (derived from lipid or polymer-based materials) ensure modest biocompatibility and biodegradability and have shown promising outcomes in clinical studies [15]. Similarly, inorganic nanoparticles (metals, silica, oxides, etc.) have also demonstrated desirable properties, including high chemical stability, ease of synthesis, and functionalization. Additionally, inorganic nanoparticles, including iron oxide nanoparticles, can be utilized as contrast agents in different imaging techniques, such as myocardial perfusion scans and magnetic resonance imaging [16]. Despite these attractive attributes, inorganic nanoparticles are less successful in clinical translation because of their poor biocompatibility and toxicity concerns [17]. 

Nanomedicine facilitates the loading of a range of active payloads, including growth factors [18], oligonucleotides [10], cytokines [11], and, essentially, chemotherapeutics [19]. The surface modification of cargo-loaded nanoparticles by targeting ligands/moieties can further improve their uptake by the tumor cells overexpressing specific receptors in an in vivo environment [20]. For example, the upregulation of folate receptors is reported in several tumor types in in vitro cell culture models and in in vivo studies involving human biopsies or experimental animal models [21]. Using nanocarriers for drug delivery enhances the blood circulation kinetics of either conjugated or entrapped drugs, protects the payload and helps cross a range of biological and physical barriers [22]. 

The number of nanomedicine formulations in advanced-stage clinical trials is continuously growing, and several nanomedicine trials have exhibited encouraging results, especially for tumor-targeted drug delivery [19]. Since the approval of Doxil^®^ and daunoxome^®^ in the mid-1990s to treat HIV-related Kaposi sarcoma, a significant advance has been made in developing nanomedicines for cancer treatment [23]. To date, 60 nanomedicine formulations are approved for clinical trials, such as the recently developed liposomal onivyde^®^ and vyxeos^®^ formulations targeting pancreatic cancer and acute myeloid leukemia [24]. These formulations involve anti-cancer medications embedded in the liposome to ensure an optimal delivery across the sarcolemma of target cells. Thus, the hydrophobic drug molecules can be optimally delivered in the intracellular environment of the cancer cells. Nanomedicines can also be used to treat viral infections, as evidenced by the approval of Moderna’s and Pfizer-BioNTech’s nanoparticle vaccines for the COVID-19 pandemic [25]. 

Apart from cancer treatment, novel nanomedicines are currently being explored to address the unmet medical needs of muscle disorders. However, several biological and pharmaceutical barriers impede the penetration of nanomedicine into skeletal muscles. One of the primary reasons for poor delivery to the skeletal muscles is the deposition of dense extracellular matrix (ECM) enriched with collagen and other glycoproteins, such as proteoglycans modified with highly negative sulfate moieties [26]. These proteoglycans and collagens impede hydration and promote a stiff ECM for the transport of molecules. Together, these ECM components hamper nanoparticle penetration into skeletal muscle due to electrostatic and mechanistic interactions [27]. In addition, cumulative fibrosis in advanced DMD may also present a physical barrier that blocks an efficient migration and/or differentiation of therapeutic cells in the skeletal muscle [28]. The physical barrier also prevents the development of an optimal cellular niche, which reduces the efficacy of cellular and/or molecular therapeutics [28] (Figure 1). 

The current review highlights significant advances in nanomedicine-based solutions to treat muscle disorders. We encompass the delivery of different active ingredients, including drugs, genetic information, and gene editing tools, using a nanoparticle platform (Figure 2). We have also highlighted the current achievements and proposed novel strategies to improve therapy outcomes and clinical translation. 

## 3. Nanoparticles for Drug Delivery in Muscle Dystrophies

Anti-inflammatory agents such as glucocorticoids are considered the gold standard treatment in DMD [29]. However, the chronic use of these steroids often leads to severe side effects, which can be lowered by encapsulating them in nanoparticles. For example, Turjeman et al. administered a steroid prodrug, methylprednisolone hemisuccinate (MPS)-loaded PEGylated nanoliposomes in dystrophic mice. The drug showed desirable bioactivity while demonstrating lower off-target effects, such as osteoporosis, when compared to the control treatment [29]. Conventional steroids may also be useful in priming the skeletal muscle before nanomedicines are administered to reduce dystrophy phenotypes. This is partly because the steroids reduce the production of inflammatory cytokines, such as IL-1 and TNF-alfa, which are primary drivers of ECM production and fibrosis [28,30]. Thus, a preliminary treatment with steroids to reduce fibrosis of dystrophic muscle may be desirable before the administration of nanomedicine.

For over 30 years, dystrophin-deficient *mdx* mice remain the most common disease model to study muscular dystrophies (DMD) and establish a good framework to evaluate disease pathogenesis and treatment outcomes. The *mdx* mice have a naturally occurring point mutation in exon 23 of the DMD gene that represents human DMD with remarkable similarities, especially in exercised mice [31,32]. The liposomal formulations of MPS were administered iv in *mdx* mice to evaluate their therapeutic potential. The unique vascular abnormality at the inflamed target tissue facilitated the passive targeting of these small nanodrugs (80 nm). The liposomal-MPS were mostly internalized into the diaphragm of *mdx* mice due to the leaky and abnormal vasculature following iv injection. A significant reduction in serum TGF-β and diaphragm macrophage infiltration was observed after a short-term treatment of four weeks. In a long-term study over 58 weeks, liposomal-MPS demonstrated improved muscle strength and mobility [29,33]. 

An attractive strategy to counteract MDs is drug repurposing, an adaptive technique for identifying new applications for previously approved investigational drugs outside the scope of their initial indication. For their safety profile, such repurposed drugs have already been extensively validated in various pre-clinical and clinical trials. Additionally, in some instances, formulation aspects are already developed [34]. To be therapeutically more efficacious, the active agents must reach the target muscle cells and myonuclei with high specificity and concentration. Drug repurposing must carefully be planned to use computational (e.g., gene expression, chemical structure, genotype) and experimental (target interactions) approaches to choose the right candidate for the therapy. Prior knowledge of chemical properties such as molecular size, structure, and hydrophobicity plays a crucial role in selecting the appropriate nanocarrier for a given repurposed drug [35].

The signature myogenic recovery factors, such as downregulated myoblast determination protein 1 (MyoD), can also be targeted to restore and promote muscle differentiation and regeneration [36]. Further, co-loading of two drugs, MyoD and glatiramer acetate (up regulator of anti-inflammatory cytokines), can be achieved using nanolipodendrosomes [37]. Such a synergetic combination of different medications may efficiently potentiate disease regression compared to monotherapy [38]. The drug-loaded nanoparticles can inherently reduce the clinically relevant drug administration dose, reducing the off-target side effects. Such nano-encapsulated drugs are manifold superior to their free drug counterparts in terms of efficacy of pharmacological doses. 

A nanocarrier system based on lipid nanoparticles of perfluorocarbon (PFC) can be used to deliver rapamycin, an immunosuppressant and anti-inflammatory agent that aids in restoring defective autophagy mechanisms in *mdx* mice [39,40]. After systemic injection, nanocarriers rescued the autophagy flux in the *mdx* mice, which improved muscle strength and cardiac contractile performance compared to ten-fold higher concentrations of the free dosage form. Consequently, the PFC-NP platform increased rapamycin tolerance and reduced side effects by lowering the required dosage. 

In addition to treating MDs, nanoparticle-based drug delivery has also shown promising results in treating muscle wasting due to malnutrition, advanced age, and lack of physical activity [41]. Anti-atrophic bioactive heptapeptides have previously demonstrated high efficacy in treating skeletal muscle pathologies [42,43]. However, their therapeutic potential is limited by the short half-life, hydrolytic stability, and poor distribution in body organs [44,45]. To address these limitations, a neutral, non-cytotoxic hydroxyl-terminated poly(amidoamine) (PAMAM-OH) dendrimer was used as an Ang-(1–7) carrier [46]. To compare the efficiency of this drug delivery system, the authors immobilized the lower limbs of *wild-type* mice for 14 days. Interestingly, loaded nanoparticles restored muscle strength and fiber diameters of immobilized limbs to a control level after intraperitoneal administration compared to angiotensin [46] (Figure 1). It is known that aminoglycosides (gentamycin) can suppress the stop codon of dystrophin in many in vitro and animal models. However, their poor delivery profile to muscle cells and toxicity reduces their therapeutic efficacy [47]. To this end, pegylated liposomes were used to encapsulate gentamycin, resulting in a significantly higher accumulation of these nano-gentamycin formulations in the cytoplasm and cytoplasmic membrane of myofibrils following intraperitoneal administration [48]. Overall, the pharmacokinetics of the drug was improved, and the drug-related cytotoxicity and nephrotoxicity were suppressed [48]. Despite these favorable properties using nanomaterial, gentamycin is a drug of choice only for nonsense DMD mutations. Thus, its clinical potential is limited to only a small cohort of the broad DMD patient population [49]. Thus, the off-target effects of a drug can be further reduced by increasing its receptor affinity and bioavailability, which will reduce the concentration required to achieve the desirable effects.

## 4. Nanoparticles for Gene Regulation in Muscle Dystrophies

Nanomedicine harboring genetic information also promises to recover functional gene expression of defective genes in diseased muscles (Figure 1). The use of naked therapeutic plasmids in gene therapy is limited due to their susceptibility to enzymatic degradation, high molecular weight, and anionic nature [50]. Nonreplicating recombinant Adeno-associated viruses (AAVs) are commonly employed to encapsulate plasmid DNA for gene delivery. The viral vector (AAV) carrying a shorter version of dystrophin obtained FDA/EMA approval for phase I or I/II clinical trials in DMD patients [51]. Meanwhile, FDA also approved AAV-based gene therapy (Zolgensma^®^) to treat children with spinal muscular atrophy (SMA) in 2019 [52]. However, despite the clinical progress of AVV-based delivery systems for gene therapy in skeletal muscle tissues [51], they are associated with many concerns. AAVs demonstrate low packaging capacity, poor tissue selectivity, and reduced efficiency in targeting muscle stem cells. Additionally, they present several clinical drawbacks, including potential toxicity at high doses in large animals [51].

Moreover, preexisting resistance to AAV is also prevalent in a significant proportion of the population [53], reducing the therapy’s success rate. The use of AAV-based vectors elicits a strong adaptive immune response in non-resistant patients. Thus, AAV cannot be used for the second dose when required [54]. Therefore, we must explore alternative ways to deliver genes into skeletal muscles, ideally using non-viral DDS. To this end, numerous attempts have been made, although they are still far from clinical settings. Recently, Jativa et al. used 5-polyamidoamine dendrimer (G5-PAMAM) conjugated to muscle homing peptide to introduce luciferase plasmid into C_2_C_12_ cells [55]. C_2_C_12_ is a commonly used immortalized murine pre-myoblast cell line that differentiates into myotubes under low serum conditions, thus representing a good in vitro model of myoblast differentiation into myotubes. Additionally, this cell line possesses features, such as myosin and glycogen content, that resemble human myotubes, enabling researchers to conduct muscle contraction studies in controlled conditions [56]. In another study, hyperbranched poly(ester amines) (PEAs) were employed to transfect plasmid into C_2_C_12_ cells and dystrophic muscles and exhibited very good biocompatibility due to its biodegradable nature [57]. In a different proof-of-principle study, IM injection of biodegradable polyphosphoester, poly(2-aminoethyl propylene phosphate) (PPE-EA) complexed with β-galactosidase plasmid demonstrated sustained release in mouse muscles [58]. Similarly, polyplex nanomicelles, based on poly(ethylene glycol) (PEG)-b-polycation and pDNA expressing luciferase, demonstrated excellent capacity for gene introduction to skeletal muscles after IV injection [59]. Despite such advancements, the nano-based approaches are still far from the efficient transfection of the therapeutically relevant amount of plasmid DNA in living animals. 

Alternatively, delivering a functional copy of a defective gene may be possible through mRNA. However, the poor stability of RNA molecules in vivo is a potential hindrance to the use of mRNA as a drug [60]. Conversely, the COVID-19 mRNA vaccine developed by Pfizer/BioNTech and Moderna has exhibited the therapeutic potential of delivering mRNA in vivo. However, it is not known if such a strategy can be used to treat muscle disorders. 

Another facet of gene modulation in muscle disorders is based on the suppression of dysfunctional mRNA by employing oligonucleotides, such as anti-sense oligonucleotides (ASO), small interference RNA (siRNA), and micro-RNA (miRNA) [61]. All these gene modulation molecules suppress the gene expression at either mRNA or splicing machinery levels and do not implicate any change to the genome. This collectively results in the suppression of the translation of malfunctioned proteins from dysfunctional mRNA. ASOs are single-stranded DNA or RNA oligonucleotides complementary to gene transcripts and exert their actions through RNase-H1-mediated degradation of target mRNA, miRNA inhibition, or inhibition of the splicing machinery [62]. The in vivo delivery of ASOs into skeletal muscle tissues is hampered due to endogenous nucleases and relatively low target sequence affinity. To an extent, chemical modification and the conjugation of oligonucleotides have addressed these chemical barriers. The modification increased the half-life of ASOs and reduced their degradation by exonucleases [10,63,64]. The in vivo delivery can be further improved by efficiently using nanotechnology and material science [10]. Several nano-based drug delivery techniques are currently under investigation in cell culture and experimental animal models. For example, the myoblasts from MD type I patients were treated with 2′O_Me/PS-modified ASOs conjugated with two cationic cell-penetrating peptides. However, these patients exhibited an aberrant transcript and RNA-mediated toxicity due to large trinucleotide repeat expansion within the DMPK gene. Conversely, the intracellular delivery of oligonucleotide-cell penetrating peptide complexes can reduce the formulation of nuclear aggregates [65]. ASOs can also function to promote exon skipping and to correct the frameshift mutations of the defective genes [62]. Several PMO ASOs (for instance, eteplirsen, golodirsen, and vitolarsen) are currently approved for treating DMD patients [3]. However, insufficient protein recovery and poor bioavailability in heart tissues require further investigations to improve clinical efficacy. The nanoparticle-based on tricycloDNA ASO (tcDNA_ASO) exhibited an optimal delivery in skeletal muscles, brain, and heart of DMD mice, which reduced the disease phenotype [66]. Consistent with this, an intracellular injection of nanopolymer-encapsulating ASOs for exon skipping resulted in a 3.4 times higher expression of dystrophin than nanocarrier-free ASO in *mdx* mice [67]. Additionally, the choice of nanocarrier is critical for targeting specific tissues. For example, in one study, chitosan-shelled nanobubbles were loaded with phosphorodiamidate morpholino (PMO) ASO to suppress DUX4 expression in dystrophic cells. However, the irreversible binding of chitosan-shelled microbubbles to PMO-ASO prevented the downregulation of the DUX4 gene [68]. It may be possible to silence the genes through siRNAs, which degrade target mRNA by recruiting an RNA-induced silencing complex. However, to our knowledge, a study has investigated the therapeutic potential of siRNAs in muscular dystrophies. Further studies are required to efficiently deliver oligonucleotides (ASO, siRNA, miRNA) to skeletal muscles to treat muscle disorders (Figure 1). 

## 5. Nanoparticles for Gene Editing in Muscle Dystrophies

As discussed, gene therapy relies on providing functional copies of a defective gene, while ASO therapy seeks to suppress gene expression. However, none of these methods provide a durable correction of gene mutation. The genome editing technique may be a therapeutic alternative to ASO treatment. The Nobel prize-winning technology of Clustered Regularly Interspaced Short Palindromic Repeats (CRISPR), associated with a specific DNA endonuclease protein called Cas9, has become a powerful gene editing tool that holds the potential for correcting gene mutations [69,70]. Cas9 recognizes its target by a chimeric single-guide RNA (sgRNA) that encodes a sequence complementary to a target protospacer, which is immediately followed by a short sequence called protospacer-adjacent motif (PAM) [71]. The delivery of Cas9/sgRNA ribonucleoprotein complexes via non-viral delivery systems is under investigation to further promote its therapeutic efficacy [72]. The main challenge in developing an entirely non-viral formulation is the large size of Cas9 and the susceptibility of ribonucleoprotein complexes to degradation [73]. However, Lee et al. engineered AuNP (CRISPR-GOLD) to enhance dystrophin expression by inducing homology-directed repair (HDR) in dystrophic mice. The gold nanoparticle was densely packed with thiol-terminated DNA to efficiently hybridize thiol-terminated donor DNA and facilitate its rapid release in the cytoplasm through disulfide cleavage. Cas9/sgRNA was then adsorbed onto the nanoparticles (owing to their affinity for DNA), and finally, the nanoparticle was covered with endosome-disrupting polymer PAsP (DET). CRISPR-GOLD successfully delivered both protein and nucleic acid components of the CRISPR/Cas9 system. The *mdx* mice were injected with cardiotoxin to exacerbate muscle injury, followed by the injection of CRISPR-GOLD or the controls. Promisingly, the HDR efficiency of CRISPR-GOLD was 18-fold higher than control, and overall, 5.4% of the dystrophin gene in *mdx* mice was corrected back to the wild-type gene. Additionally, the nanoparticles had no immune response, suggesting the possibility of multiple-dose administration [74]. Similarly, Wei et al. have also shown the potential of delivering Cas9/sgRNA to multiple tissues, including muscle, brain, liver, and lungs, using lipid nanoparticles (LNP). Interestingly, these nanocarriers restored dystrophin expression up to 5% in ΔEx44 DMD mice following IM injection [73]. Recently, a biodegradable nanocapsule has been used to encapsulate Cas9/sgRNA to efficiently deliver the complex to different tissues, including skeletal muscles [75]. Although CRISPR/Cas9 is a revolutionary technology in gene editing, novel drug delivery systems are required to improve precise gene correction. In addition, strategies that efficiently target proliferating satellite cells and muscle fibers are emerging issues for future nanomedicine for muscle diseases that must be addressed accordingly (Figure 1). 

## 6. Limitations in the Application of Nanoparticles

The applications of nanomedicine to muscle dystrophies are still in their nascent stages; however, the recent encouraging results from in vitro and pre-clinical studies are opening new doors for clinical translation. Nevertheless, the gap between in vitro and in vivo testing must be filled to discover the potential of nanomedicine application to MDs treatment. Currently, our extensive knowledge and understanding of nanomedicine are mostly limited to cancer therapy and vaccine development. Even in the most mature cancer nanomedicine discipline, challenges remain in their clinical translation despite the attractive attributes embodied by nanotechnologies. The success rates for nano-drug potential candidates for Phases I, II, and III clinical trials significantly plunge from 94% to 48% to 14%, respectively [76]. A large body of literature supports different nano-based therapeutics, yet the FDA has approved only a handful. The inequity of academic output and translation success originates from several challenges, for example, the complexity of nano-assemblies of different components, rendering it difficult to identify their unclear cytotoxicity profiles. Additionally, nano-bio interactions (i.e., body fluids, ECM, and cellular components) have not been studied in clinically relevant model systems. Once they enter the circulatory system, nanoparticles are immediately coated by a “protein/ biomolecular corona”, which consists of lipids, proteins, and metabolomes (Figure 3). 

By altering the synthetic identity of the nanoparticles, this biomolecular corona significantly affects their pharmacokinetics, biodistribution, and target capability and ultimately defines their fate in vivo [77]. The reproducibility and large-scale synthesis of nanoparticles with distinct properties perpetuate another challenge per se. Besides the substantial scientific expertise, instrumentation and characterization methods that might lead to a new drug product development are also relatively high [78]. Another concerning reason is the lack of suitable biomarkers for nanomedicine, particularly if they are not targeted to a specific receptor(s) (Figure 2). Non-invasive imaging modalities such as PET and SPECT seem to be good starters for patient stratification, but these technologies are devoid of cost and time efficiency [79]. Apart from that, the contribution of gender must also be considered a determinant of nanomedicine evaluation, as suggested by the recent analysis of the effectiveness of COVID-19 vaccines developed by Pfizer/BioNTech and Moderna. Based on the available data, both vaccines showed a (slightly) better effectiveness in males than in females [80]. Biological barriers in MDs are embodied by the complex architecture of the skeletal muscles, which is enriched with dense extracellular matrix (ECM), accounting for 1–10% of muscle mass [81,82]. The fibrous proteins in ECM (collagen, glycoproteins, proteoglycans, etc.) impede nanoparticle penetration via electrostatic and mechanical interactions [27,83] (Figure 2).

An analogous ECM barrier is also found in the tumor microenvironment, which represents a formidable barrier circumventing the intra-tumor drug delivery and therapeutic efficacy of many anti-cancer nanotherapies [79,84,85]. It would be highly recommended to pay more attention to the crucial physical, biological, and pathophysiological factors that might accelerate a successful clinical translation from bench to bedside.

## 7. Future Perspectives

The aberrant ECM deposition in damaged muscle fibers impedes the penetration of nanoparticles to reach their target muscle cells. Transport of the nanoparticles through ECM is more complicated due to its inherent mesh-like organization of roughly 10 nm to hundreds of nanometers. These pores exclude/reject larger nanoparticles due to frictional interactions, electrostatic interactions, or steric hindrance and ultimately hinder the rapid and uniform penetration of larger nanoparticles. Notably, the pore size in ECM may vary from one pathological condition to another [86,87]. In this pursuit, choosing the right nanoparticle size might be the key to effective penetration (Figure 2). A balance must be investigated among sizes of nanoparticles, as small nanoparticles are cleared away more rapidly (renal clearance < 10 nm). In contrast, it is difficult for larger nanoparticles to penetrate and prone in the reticuloendothelial system (RES), mostly larger than >200 nm. Similarly, the remodeling of ECM might be a suitable target to facilitate deep nanomedicine penetration to the target cells (Figure 2). The ECM can be modified by promoting its degradation or reducing its synthesis by inhibiting Tumor-Associated Fibroblast (TAF) activity. For example, in the context of the tumor microenvironment, Tang et al. developed dual-functional bromelain-immobilized and lactobionic acid (LA)-functionalized chitosan nanoparticles encapsulating doxorubicin, in which bromelain was responsible for ECM degradation, and LA further increased the accumulation of the nanoparticles via active tumor-targeting [88]. Similarly, the overexpression of components of the ECM in pathological states can be significantly reduced by blocking the growth factors involved in signaling for TAF stimulation. For example, the inhibition of TGF-β signaling with antibodies reduces collagen synthesis and promotes nanomedicine delivery in animal models. Thus, the treatment of experimental animals with a TGF-β neutralizing antibody increased the tissue delivery of Doxil and exhibited superior therapeutic efficiency than the control group [89]. Generally speaking, the modification of ECM primarily benefits the delivery of larger nanomedicine [90], as they are more hindered by ECM [91]. Thus, the treatments targeting ECM content may be employed to reduce the physical barrier before appropriate nanomedicines are administered for treating muscular dystrophies. We believe that lessons learned from cancer nanomedicine and the perspective strategies could be far more effective than those available at present. 

The high specificity of the antibody to the corresponding antigen can be exploited to achieve active nanoparticles targeting skeletal muscle cells. However, no scientific study has been conducted on antibody-functionalized and muscle-targeted nanoparticles [92]. An alternative to antibodies, short peptide sequences can be surface decorated on nanoparticles to promote active targeting and penetration. For example, RGD peptide (Arginine-Glycine-Aspartate) can prime the integrin αvβ3 family to ensure a targeted delivery to tumor vasculature and tumor cells. As observed in a study by Li et al., modified RGD-conjugated nanoparticles facilitate in vivo tumor regression [93]. A highly potent αvβ3 ligand, i.e., cyclic RGD-tyrosine-lysine peptide (cRGDyk)-modified liposome encapsulating cisplatin, improved therapeutic efficacy against bone metastasis in clinical trials [94]. Polymeric nano-systems have been functionalized with agents that selectively bind active molecules or cell surface receptors expressed on muscle fibers. Active targeting-dependent uptake has been demonstrated using PLGA nanocarriers functionalized with a muscle-homing peptide M12 [95]. The exploration of additional molecular targets in muscle dystrophies and engineering cognate nanoparticles will further facilitate the nanomedicine delivery to muscle cells (Figure 2). For instance, the overexpression of TLR7 is prevalent in dystrophin-deficient skeletal muscle, which offers a molecular target via a cognate peptide design [96]. Similarly, nanoparticles can be surface decorated with cell-penetrating peptides (CPPs) to promote cellular entry of nanoparticles to the target cells. In a study, recombinant dystrophin R16/17 protein was fused to a CPP (mTAT), facilitating muscle penetration and restoring sarcolemma nNOS delocalization [97]. The use of in silico rational drug designs comprising molecular docking and structure-activity relationship studies can further narrow down the lead candidate screening process [98,99]. 

Macrophage polarization can be another attractive therapeutic target for treating muscular dystrophies (Figure 2). Macrophages are abundantly found in skeletal muscle during regeneration. The M2 macrophages have anti-inflammatory properties and promote muscle differentiation and repair following injury. Conversely, an imbalance of M1 to M2 polarization impairs skeletal muscle repair [100]. Some inorganic nanoparticles, such as nanoceria, gold, and titanium oxide, promote immunomodulation through macrophage polarization [101]. Although their precise roles in dystrophic skeletal muscle remain elusive, these nano-systems may have therapeutic potential in treating skeletal muscles with aberrant inflammation.

The recent understanding of the pathogenic mechanism of MDs highlights the urge to encompass novel and improved remedies. For instance, pentamidine (PTM)-loaded nanomedicines were used to investigate the off-target effects of the drugs to reduce toxicity. Ongoing studies are aimed at demonstrating the efficacy of this novel formulation in treating MD1 (in progress). However, several previous research articles on PTM-loaded nanocarriers only describe preliminary in vitro results or formulation studies. Therefore, a deeper analysis of the in vivo behavior of the PTM-loaded nanocarrier is required. Particularly using suitable animal models for which PTM has been repurposed [102,103].

Peering at the last few decades’ progress of cancer nanomedicine, poor clinical translation has been a bottleneck. Therefore, the therapeutic efficacy of nanomedicine can be improved by using a focused design and a decision-making strategy. Moreover, the ongoing and future trials warrant the need for developing, validating, and implementing patient pre-selection tools to obtain the optimal efficacy of nanomedicines. The implementation of companion diagnostics to identify patients likely to benefit from the therapeutic version can be a baseline for patient pre-selection/stratification. Continuous development in imageable nanoparticles, state-of-the-art high-throughput imaging technologies, and advances in bioanalytical methods for drug and nanomedicine visualization and quantification might improve their clinical translation over time [104].

## Figures and Tables

**Figure 1 ijms-23-12039-f001:**
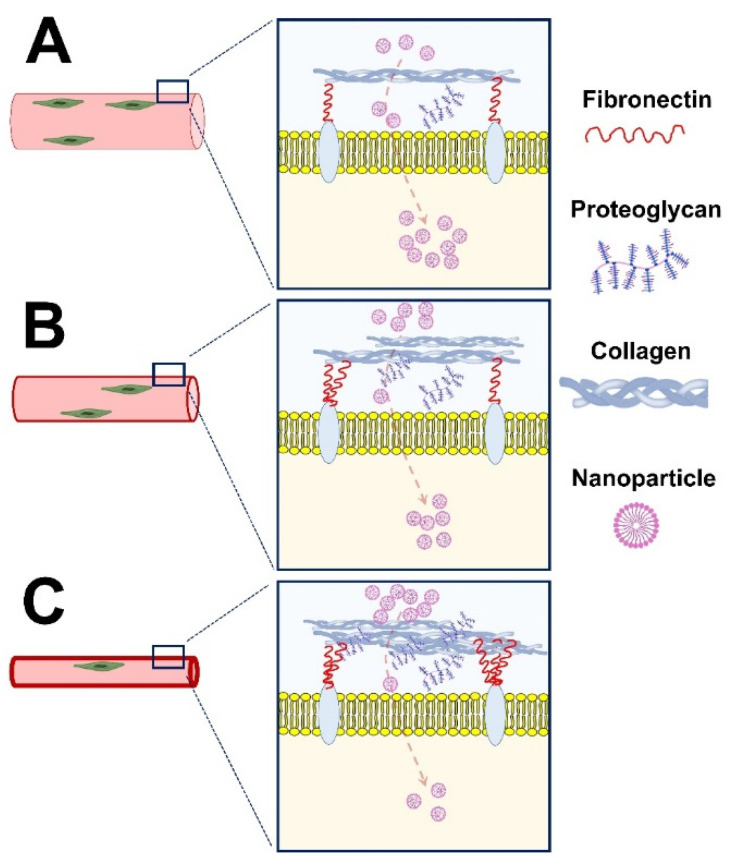
The penetration of nanoparticles in healthy (**A**) and dystrophic skeletal muscle with early (**B**) and advanced (**C**) dystrophy. The amount of extracellular matrix, including fibronectin, proteoglycans, and collagens, is a critical driver of nanoparticle penetration into skeletal muscle. The payload delivery and treatment success rates are higher in skeletal muscles with early vs. advanced dystrophy.

**Figure 2 ijms-23-12039-f002:**
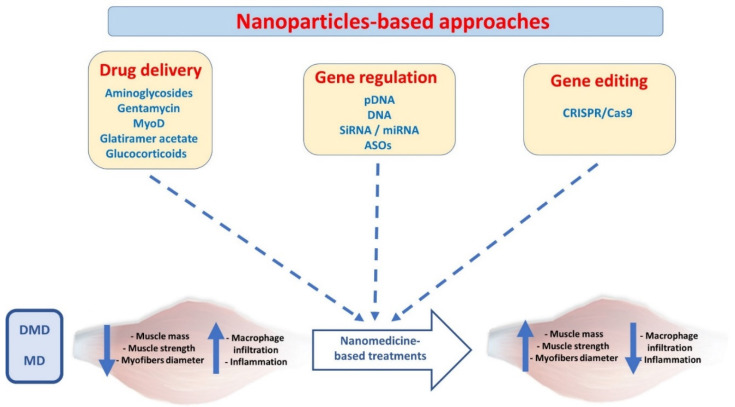
Nanoparticle-based approaches to treat skeletal muscle dystrophies. (MyoD; myoblast determination protein 1, pDNA; plasmid DNA, siRNA; small interference RNA, miRNA; micro-RNA, ASO; anti-sense oligonucleotides, DMD; Duchenne muscular dystrophy, MD; myotonic dystrophy).

**Figure 3 ijms-23-12039-f003:**
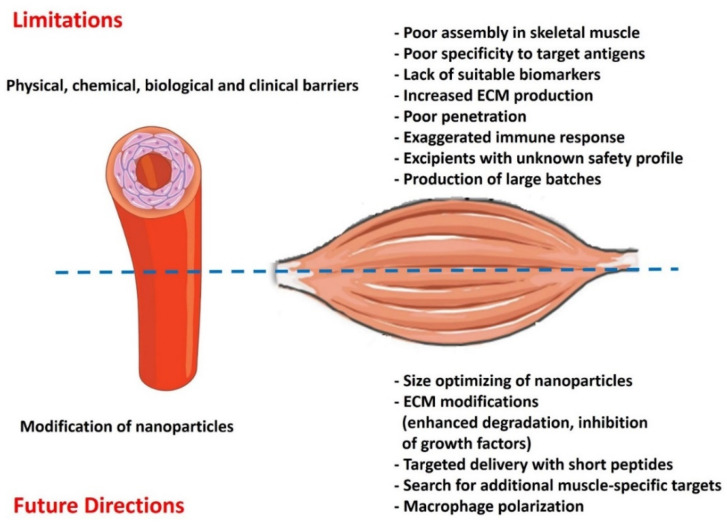
Potential limitations and future directions for using nanomedicine in treating muscular dystrophies.

## Data Availability

Not applicable.

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
