# Peer review of "Nanomedicine for Treating Muscle Dystrophies: Opportunities, Challenges, and Future Perspectives"

_ijms, 2022, doi:10.3390/ijms231912039_

Round 1

Reviewer 1 Report

Comments to be addressed by the authors

Title

Line 9: the semicolon following the email address should not be bolded or italicised.

Line 11: change “UAE” to “27272, United Arab Emirates” for consistency with the above addresses

Abstract

Requires strengthening of terminology and a more descriptive account of the diseases.  The authors should attempt to include the following:

Impaired skeletal muscle regeneration, leading to pathological inflammation that drives muscle wasting, leading to weakness, functional dependency and premature death.

Causes of death include respiratory failure due to diaphragm muscle decay.

That conventional therapies aim at offsets/ameliorates symptoms is not entirely accurate.  The most common therapies are treatments with varied classes of glucocorticoids to suppress inflammation; whilst this offsets/ameliorates symptoms, it also promotes a physiological rather than pathological inflammatory environment, which, in-turn, promotes physiological skeletal muscle regeneration. 

A more accurate description would be something like, “conventional therapies aim to ameliorate muscle wasting by promoting physiological muscle regeneration and growth.  However their effect on muscle function remain limited, illustrating the requirement for major advancements in novel approaches to treatment such as nanomedicine.”

This will provide the readers in the related fields with a more accurate overview of that which will be discussed in further detail in the artice.

Minor suggested edits

Line 14: “ameliorating” would be a more appropriate description than “offsetting”

Line 14: “definitive” would be more appropriate than definite;

Introduction

The introduction is well-structured, informative and effective at narrowing the scope of the Review to nanomedicine.

Minor suggested edits

Line 37: "After exhausting the its regenerative capacity, the skeletal muscle develops…"

Line 40: "which leads to premature [insert: “death”]  due to dysfunction of respiratory muscles"

Line 42: "Currently, no therapeutic interventions do not have the efficacy required to inhibit or reverse the pathological end-points inis available to offset muscular dystrophies."

Line 44: "the disease to improve quality of life the lifestyle [1]"

Line 45: "rapid clearance from the body prevent [insert “promising”] clinical trials"

2. What is nanomedicine, its history, and applications

Line 63: Either place a question mark at the end of the heading or reword the heading so as not to be a question.

Line 73: What is the threshold for “extensively utilised”? I think this is an overstatement, especially in the context where the authors go on to state, "modest biocompatibility and biodegradability and have shown promising outcomes in clinical studies […] inorganic nanoparticles are less successful in clinical translation because of their poor biocompatibility and toxicity concerns etc." (Lines 75–82.); and

"To date, 60 nanomed-94 icine formulations have received clinical approval and entered the market, such as the 95 recently developed liposomal onivyde® and vyxeos® formulations [25]" (Lines 94–96.)

Lines 85–86:  Could the authors shed some clarification on what is meant by, “…by the cells overexpressing specific receptors"? 

Specifically,

·       This statement is too generic – are the cells in vitro, in vivo, ex vivo, etc?

·       Are the receptors complementary to the payload/s delivered?

·       Is the over-expression due to the targeted pathology or is it a result of an in vitro and/or animal model/s engineered to express the receptors at higher levels?  Nobody knows until they check the references, which is annoying when you just want a "good read".

Line 96: Can the authors please insert additional information to the effect of, “…recently developed liposomal onivyde® and vyxeos® formulations [insert: “respectively targeting pancreatic cancer and acute myeloid leukaemia”] [25]" (Thus providing more fulsome examples of the preceding statement vis-à-vis development for cancer treatment.)

Interim general comment:

The Review article will receive interest from, amongst others:

            ·       the muscular dystrophies’ field; and

·       the nanomedicine field, and

·       the muscular dystrophies and/or other muscle disorders’ field who are also familiar with the nanomedicine field. 

Researchers in the muscular dystrophies’ fields, for example, may not know or even heard of “onivyde® and vyxeos®”.  The authors need to complete a 'sweep; of their article from the perspective of the readers in the audience and ensure to clarify information that may not be apparent to other highly relevant research fields.

Line 103: Understanding that this is introducing the next section, it would be apt to mention in addition to collagen and glycoproteins, “proteoglycans modified with highly negatively charged sulphate moieties” (there should be a handful of articles in the past 5 years highlighting that component of the ECM in dystrophic muscles for the authors to choose from).

Further to this, can the authors consider introducing the concept of a “stiff ECM” in fibrotic muscles that presents an even more challenging environment by way of a physical barrier in cases of late treatment?

Line 114: ASO should be defined as “anti-sense oligonucleotides”

3. Nanoparticles for drug delivery in muscle dystrophies

This section of the Review could be better structured. 

Whilst the content is sound and relevant, I would prefer to see this section to:

·       begin with acknowledging that glucocorticoids are still the gold standard treatment;

·       introducing the mdx mice;

·       bring lines 154–164 to the forefront;

·       then introduce drug repurposing, but lines 117–128 only;

·       then continue with lines 138 onwards following the theme of repurposing anti-inflammatory payloads and co-delivery potential;

·       move lines 128–137 down, creating a natural progression away from anti-inflammatory targeting to strategies that affect gene expression (which will prime the reader for the next section, targeting gene expression).

(See further, the suggested restructuring of this section at the end of this review.)

Line 122: Consider substituting “more effective” with “efficacious” (ie. “To be therapeutically more effective efficacious, the active agents must…” (see line 130 where this terminology is used astutely).

Lines 123 – 141: Comment for the authors’ consideration, to be actioned at the authors’ discretion

·       Arguably, a highly specific drug will have drug-target molecule (eg. receptor) affinity in the nanomolar range such that lower concentrations of the payload(s) is/are required.

·       This is consistent with that which the authors state at lines 143 – 146: “The drug-loaded nanoparticles can inherently reduce the clinically relevant drug administration dose, reducing the off-target side effects. Such nano-encapsulated drugs are manifold superior to their free drug counterparts in terms of efficacy of pharmacological doses.”)

·       Given the challenging environment of skeletal muscle’s ECM, the authors could discuss the idea of “priming” the muscle ECM with conventional high efficacy glucocorticoid therapy that reduces inflammation and hence the ‘over-deposition’ of ECM molecules, chief amongst them being collagen, whose expression are driven by inflammatory cytokines, IL-1, TNF-alpha etc.  Hence preserving a, well hydrated ECM through which molecules can diffuse relatively unimpeded. 

·       The glucocorticoids can be withdrawn before they start to become detrimental. This would presumably offer a healthier ECM for both penetration and delivery and glatiramer acetate may be even more efficacious if it is delivered into an environment with its inflammation already dampened.  I note at lines 140  41: the authors discuss a more refined strategy of the co-loading of two molecules “...MyoD and glatiramer acetate (up regulator of anti-inflammatory cytokines) can be achieved using nanolipodendrosomes [34].” This would also fit the above hypothesis, although it is not “priming”, which patients may ultimately require depending on their DMD’s progressive state.

·       Highly negatively charged chondroitin sulphate moieties on chondroitin sulphate proteoglycans a (CSPGs) (and other PGs such as those modified with dermatan sulphate) are responsible for drawing water molecules into the ECM (due to the overall net positive charge of water); however, in the case of fibrosis, collagens are deposited onto the CSPGs, which act as a “transitional matrix” impeding hydration and promoting a stiff ECM through which not many molecules can penetrate. 

·       Aside from intracrine signalling, every cell signalling event is mediated by the ECM – i.e., every cytokine, growth factor, peptide hormone, etc. must encounter the ECM before it hits its cell receptor (and the same is true for lipophilic molecules such as steroid hormones, etc; before diffusing across the phospholipid bilayer, also must encounter the ECM.

***

Review continued:

The authors introduce “mdx” mice without any background as to what these mice are a model of.  Please refer to the below suggested restructuring of this section in which the following is requested of the authors:

[Reviewer’s comment: insert something to the effect of, “Proof of principal and preclinical studies, including with conventional and next-generation glucocorticoids have exploited the mdx mouse model of DMD.  Mdx mice have a naturally occurring point mutation in exon 23 of the dmd gene that represents human DMD with remarkable similarities, especially in exercised mice.]  (Please cite:  Swiderski K, Lynch GS. Murine models of Duchenne muscular dystrophy: is there a best model? Am J Physiol Cell Physiol. 2021 Aug 1;321(2):C409-C412. doi: 10.1152/ajpcell.00212.2021. Epub 2021 Jul 14. PMID: 34260298.) 

4. Nanoparticles for gene regulation in muscle dystrophies

Line 182: define AAVs for the first time it is used, i.e. “Adeno-associated virus (AAV)…”

Line 183: please use another term other than “got”, such as “obtained”

Line 198: the authors introduce C2C12 cells with no background information.  Please insert a short description of what they are and how they differentiate into myotubes in culture, hence providing a relatively good in vitro model of myoblast differentiation into myotubes.

Line 212: please change, “…has been tampering with…” with “…has been a hindrance to…”

Lines 216–221:

1.       please use different terminology than “malfunctioned gene”, such as “dysfunctional mRNA”

2.       please amend this paragraph to describe that more accurately that ASOs, siRNAs and miRNAs are not supressing “the genes”, but rather, they are supressing (or regulating the splicing of) the gene product, which is dysfunctional mRNA.  This, in-turn, supresses the translation of “malfunctional protein(s)” from “the dysfunctional mRNA”.  In the case of regulating the splice sites of the mRNA, this is promoting translation of functional protein.

Line 225:  please amend ASO to ASOs.

Lines 225 – 226: Please amend: “To further improve in-vivo delivery, nanotechnology, and material science promise to overcome biological barriers intracellular delivery [19].” to read, “To further improve in-vivo delivery, the use of nanotechnology, and material science to overcome biological barriers and enhance intracellular delivery are promising approaches [19].”

Line 228: please amend: “In a study…” to “In one study…” or "[insert author] et al showed"

Lines 221 to 222: Please amend, “…within DMPK gene leads to an aberrant transcript…” to read, “…within the DMPK gene led to an aberrant transcript…”

Lines 224 to 225: Please amend, “ASOs can also implicate their function through exon skipping…” to “ASOs can also function to promote exon skipping…”

Line 249: See line 228, above.

5. Nanoparticles for gene editing in muscle dystrophies

This section is well-written with an astute synthesis of complex information, including the underlying mechanisms of CRSPR technology.  There are no recommended changes.

6. Limitations in the application of nanoparticles

Line 299: the term is “opening new doors”, not “…new gates”

Line 300: please change, “disclose” to, “discover”

Line 311: please change, “proper” to, “relevant” (or, “most relevant”)

Lines 311–312: please amend, “Once exposed to the bloodstream, nanoparticles are…” to, “Once they enter the circulatory system, nanoparticles are…” (or, “Once in circulation, nanoparticles are…”

7. Future perspectives

Line 354: please define “TAF” (I assume this means, “tumor-associated fibroblast”?)

Lines 359 – 360: please amend, “ECM deregulation can be realized by blocking the growth factors involved in signaling for TAF stimulation.” to, “The over-expression of components of the ECM in pathological states can be significantly reduced by blocking the growth factors involved in signaling for TAF stimulation.”

A comment on the Figures

This manuscript requires an astute Figure

The existing figures whilst informative and technically correct, are very simplistic.  What makes a good review great is an excellently detailed and well-thought out figure.

The authors have the liberty of the figure they wish to create; but in my opinion, a figure showing healthy and compromised muscle fibres (and the stem cells) embedded into a network of ECM molecules (showing the actual molecules, ie. collagen, fibronectin, proteoglycans such as versican, SLRPs, etc. See https://www.frontiersin.org/articles/10.3389/fphys.2020.00253/full for ideas).  Then build into the figure nanoparticles carrying and delivering payloads of choice - show on one side severe fibrotic ECM that nanoparticles become trapped in; and on the other side healthy ECM that nanoparticles can move through freely and converge the two sides into the middle where there is moderate damage but still enough availability for nanoparticle penetration.  Comment in your Figure legend that the earlier the treatment, the better the success rate and that there is a threshold of nanoparticle mediated delivery of payloads.

Author Response

The rebuttal letter to reviewer 1 is uploaded below

Reviewer 2 Report

Presented review by Ahmad and Qaisar „Nanomedicine for treating muscle dystrophies: opportunities, challenges and future perspectives“ covers various approaches to nanomedicine in the field of muscular dystrophy; drug delivery, gene regulation, CRISPR DNA editing. The review does an excellent job in summarizing the current knowledge about the available nanotechnologies, as well as summarization of the specific aspects of muscular dystrophies and the concerns connected to the nanotherapy strategies. The review is well divided in separate chapters focusing on the overview of historical and current state of nanomedicine, before it switches focus to drug delivery, gene regulation and gene editing. The text well descripes the advantages and disadvanteages of the therapies in the limitations chapter and adresses the limitations of the specific approaches as well as perspectives for improvements. The text is easy to read and understand, even with extensive library of sources (101 references). The figures summarize the text in simple, but effective way. I have no major concerns, only few minor notes to be improved:

Use of unexplained abbreviations should be avoided – DMPK and CNBP, line 35

Line 40, I believe the line was supposed to be ...“premature death due to dysfunction...“

Line 55-57 – sentence is unclear

Author Response

The rebuttal letter to reviewer 2 is uploaded below

Round 2

Reviewer 1 Report

Overview of the revised manuscript

The authors have done an exceptional job in turning this manuscript around with significant improvements, chief among them being the inclusion of Figure 1.

Minor revisions:

I highlight the nature of the minor revisions required as follows:

Reference(s) to manuscript

Line 43: 'Drug metabolism and pharmacokinetics (DMPK) or Cellular nucleic acid-binding protein 43 (CNBP) CNBP genes'

Reviewer's comment(s)

The terms do not require capitalization of the first word unless it is at the beginning of a sentence.

Reference(s) to manuscript

Lines 98 to 99

Reviewer's comment(s)

This 'in vivo' should be italicized.

Reference(s) to manuscript

The entire manuscript

Reviewer's comment(s)

Please carefully proof-read the manuscript with a focus on minor discrepancies/inconsistencies of the same nature as those exemplified above.

I do not need to see the manuscript again.

***

Author Response

The response to reviewer is attached below 
